# Screening for Low Energy Availability in Male Athletes: Attempted Validation of LEAM-Q

**DOI:** 10.3390/nu14091873

**Published:** 2022-04-29

**Authors:** Bronwen Lundy, Monica K. Torstveit, Thomas B. Stenqvist, Louise M. Burke, Ina Garthe, Gary J. Slater, Christian Ritz, Anna K. Melin

**Affiliations:** 1Rowing Australia, Canberra, ACT 2600, Australia; blundy@rowingaustralia.com.au; 2Mary MacKillop Institute of Health Research, Australian Catholic University, Melbourne, VIC 3000, Australia; 3Department of Sport Science and Physical Education, Faculty of Health and Sport Science, University of Agder, 4630 Kristiansand, Norway; monica.k.torstveit@uia.no (M.K.T.); thomas.b.stenqvist@uia.no (T.B.S.); 4Norwegian Olympic Sports Centre, Department of Sports Nutrition, 0806 Oslo, Norway; ina.garthe@olympiatoppen.no; 5School of Health and Behavioural Sciences, University of the Sunshine Coast, Sippy Downs, QLD 4556, Australia; gslater@usc.edu.au; 6National Institute of Public Health—SDU, 1455 Copenhagen, Denmark; ritz@sdu.dk; 7Department of Sport Science, Faculty of Social Sciences, Linnaeus University, 351 95 Vaxjo, Sweden; anna.melin@lnu.se

**Keywords:** testosterone, endurance, questionnaire, validation, EHMC

## Abstract

A questionnaire-based screening tool for male athletes at risk of low energy availability (LEA) could facilitate both research and clinical practice. The present options rely on proxies for LEA such screening tools for disordered eating, exercise dependence, or those validated in female athlete populations. in which the female-specific sections are excluded. To overcome these limitations and support progress in understanding LEA in males, centres in Australia, Norway, Denmark, and Sweden collaborated to develop a screening tool (LEAM-Q) based on clinical investigations of elite and sub-elite male athletes from multiple countries and ethnicities, and a variety of endurance and weight-sensitive sports. A bank of questions was developed from previously validated questionnaires and expert opinion on various clinical markers of LEA in athletic or eating disorder populations, dizziness, thermoregulation, gastrointestinal symptoms, injury, illness, wellbeing, recovery, sleep and sex drive. The validation process covered reliability, content validity, a multivariate analysis of associations between variable responses and clinical markers, and Receiver Operating Characteristics (ROC) curve analysis of variables, with the inclusion threshold being set at 60% sensitivity. Comparison of the scores of the retained questionnaire variables between subjects classified as cases or controls based on clinical markers of LEA revealed an internal consistency and reliability of 0.71. Scores for sleep and thermoregulation were not associated with any clinical marker and were excluded from any further analysis. Of the remaining variables, dizziness, illness, fatigue, and sex drive had sufficient sensitivity to be retained in the questionnaire, but only low sex drive was able to distinguish between LEA cases and controls and was associated with perturbations in key clinical markers and questionnaire responses. In summary, in this large and international cohort, low sex drive was the most effective self-reported symptom in identifying male athletes requiring further clinical assessment for LEA.

## 1. Introduction

Awareness and understanding of the impacts of low energy availability (LEA) in athlete populations has continued to evolve and stimulate research interest. Energy availability (EA) is defined as the amount of dietary energy remaining for all other metabolic processes after the energy cost of exercise has been subtracted [1]. Short term (5 days) clinical studies in eumenorrheic women have demonstrated that the pulsatility of the luteinising hormone is disrupted when they are exposed to an EA below 126 kJ (30 kcal)/kg fat free mass (FFM)/day [2], although a specific EA threshold below which menstrual disturbances are induced is not supported [3]. While the interplay of LEA with bone health and menstrual function in female athletes is relatively well understood [4], an equivalent understanding in male athletes is still developing. The concept of Relative Energy Deficiency in Sport (RED-S) and the Male Athlete Triad [5] has encouraged researchers and clinicians to explore LEA in both male and female athletes and to look for a broader range of potential consequences [6,7,8].

Prevalence of LEA in male athletes is relatively undescribed, with early estimates ranging between 25 and 70% in road cyclists, distance and cross-country runners and jockeys [9,10,11,12,13]. Few studies have induced LEA in males in a controlled setting [14,15,16,17], instituting LEA at 62 kJ (15 kcal)/kg FFM for a period of 4–6 days and with a limited scope of investigation such as bone or iron metabolism markers. These short periods and thresholds of LEA have not been reflective of perturbations seen with females at a similar level [2], with cross-sectional field studies or severe energy restriction research in males [18]. It is possible males have a higher tolerance to LEA severity and/or duration and gender-specific thresholds are required. It has been noted that male endurance athletes may have chronic lower testosterone levels (40–75% of normal, healthy, age-matched sedentary males), a condition described as the Exercise Hypogonadal Male Condition (EHMC) [19], but whether this is a normal adaptation to training or due to LEA is still under debate [20]. Causation appears to parallel that in female athletic populations with identified contributors, including disordered eating behaviour [21,22,23], exercise dependence [24] and participation in aesthetic, weight sensitive or endurance sports [25,26,27].

The causes of LEA are multi-factorial and include a misunderstanding of the energy needs for sport, limitations of food availability, dietary restraint, and overzealous weight loss programs (including excessive amounts of exercise), and disordered eating or eating disorders [6]. Regardless of origin, LEA can act as a serious impediment to good health and sport performance [28,29,30]. Indeed, there is increasing evidence that exposure to LEA in male athletes is associated with effects on the hypothalamic-pituitary-gonadal (HPG) axis [9,31,32,33,34,35,36,37,38,39,40,41], changes to immune function [39,42] impairments of bone health [43,44,45] and reproductive function [46], and negative outcomes for performance [10,42] and body composition [47].

These limitations aside, the identification and appropriate management of LEA is a core competency for practitioners who work with athletic populations. A quantitative assessment of EA from measurements of energy intake, exercise energy expenditure and fat free mass is time consuming and impractical as a broad-scale screening tool for use by clinicians. Of greater importance, such assessments are fraught with potential errors or misrepresentation [48], making research in this area more challenging. Surrogate markers of EA may provide alternative ways to assess athletes for risk of the health and performance consequences of LEA [49]. The measurement of resting metabolic rate (RMR) is an accepted method [50,51,52] but requires technical skill and equipment and is also impractical on a large scale. Low bone mineral density (BMD) is often seen in those presenting with LEA (reviewed in [53]) but may not differentiate between current LEA and previous exposure that may have been resolved. Blood markers, including changes to hormones such as testosterone, insulin-like growth factor-1 (IGF-1), triiodothyronine (T_3_), insulin, blood lipids, leptin, and cortisol have been associated with LEA in males [14,18,32,54,55,56,57,58] but are beyond the budget for most sport organisations, teams or clubs to use routinely. Given this, there is interest in the development of a screening tool that could help triage those male athletes requiring specific follow-up to investigate LEA.

Screening questionnaires provide a framework to assess groups to identify those at risk and requiring further follow up. The Low Energy Availability among Females Questionnaire (LEAF-Q) is a screening questionnaire for LEA which was developed in a female endurance athlete population [59]. It provides an opportunity to triage a larger group of athletes to identify those requiring further follow up or as a simple way to track changes in individuals or groups over time. Since publication, this questionnaire has been used clinically and in research settings to assess prevalence of LEA risk and consequences in different populations [60,61,62,63,64,65] and has encouraged awareness and further research in this area.

A variety of approaches have been used in male athletic populations as a proxy for clinical identification of LEA, such as the exercise dependence scale (ExDS) [24] or eating disorder questionnaires, such as the eating disorder examination questionnaire (EDE-Q) [66]. The male and female athlete triad coalition have recommended a series of questions to screen for the male athlete triad along with a cumulative risk assessment (CRA) tool adapted from females, excluding the menstrual cycle questions [5]. Whilst these questions have good scientific logic, they are intended to identify bone health and eating disorder risk rather than LEA, per se, and have not been validated for this purpose. This modified CRA has been used successfully to assess the risk of bone stress injury [25]. Others have used the LEAF-Q with the menstrual function section removed and scores adjusted to allow for the lower number of questions [67] or replacing the menstrual function questions with those around sex drive and morning erections [66]. In a large-scale study by Hackney et al., a combination of validated questionnaires regarding physical characteristics, training and sex drive demonstrated that higher training loads are predictive of lower sex drive; however, EA was not considered [68]. Similarly, the Androgen Deficiency in Aging Males questionnaire (ADAM-Q) [69] has been used to identify male athletes with changes to their reproductive function [70], but it is unclear whether the symptoms identified are due to LEA or other causes such as chronic endurance training [68]. The Sport Specific Energy Availability Questionnaire and Interview (SEAQ-I) [10] is a questionnaire and clinical interview developed for male cyclists but relies on practitioner expertise for use and has been assessed for content validity only. It assumes LEA based on reported energy restriction and weight change. The validation process for the RED-S Specific Screening Tool (RST) [71] was inadequate, correlating scores against the pre-participation gynaecological examination [72], which itself has not been validated and was developed for adolescent females and without sufficient attention to sex differences in presentation of LEA symptoms. The Dance Specific Energy Availability Questionnaire (DEAQ) [26] utilizes questions from previously validated questionnaires including LEAF-Q and ADAM-Q [69], as well as questions used in the RED-S Clinical Assessment Tool (RED-S CAT) [73] and SEAQ-I [10]; however, these have not been validated to identify LEA in male athletic populations, either separately or in the current format.

In summary, despite the obvious interest and need for both clinicians and researchers [7,66], a validated questionnaire that could be used as a screening tool for LEA in male athletes does not currently exist. Accordingly, this study aimed to use clinical markers associated with LEA in males to develop and validate a screening tool, the Low Energy Availability among Males Questionnaire (LEAM-Q) for adult sub-elite to elite male athletes.

## 2. Materials and Methods

A total of 405 male athletes were recruited in a multi-centre study, undertaken as a collaboration between the Australian Institute of Sport, the Norwegian Olympic and Paralympic Committee and Confederation of Sports, the University of Copenhagen and the University of Agder. Inclusion criteria were elite and sub-elite male athletes, 18–50 years old with an absence of thyroid or metabolic disease. All subjects received information regarding the background of the study, test procedures and signed an informed consent document. Ethics approvals for each testing site were granted by the Australian Institute of Sport Ethics Committee, the Capital Region of Denmark, the University of Agder’s Faculty Ethics Committee, the Norwegian Regional Committees for Medical and Health Research Ethics and the Norwegian Centre for Research Data (NSD). The questionnaire was created using content from the LEAF-Q [59], ADAM-Q [74], REST-Q [75], literature review and expert consultation for content validity. Each question was scored on a Likert-type ordinal or nominal scales, with a higher score indicating a greater likelihood of LEA.

The validation was assessed in a two-step process, first for internal consistency and reliability in a young adult male athlete population (*n* = 53) and secondly, in a separate participant group, described below, to verify the self-reported symptoms from the questionnaire against measured clinical markers associated with LEA (*n* = 352). The questionnaire initially included 33 items covering dizziness, gastrointestinal function, injury and illness and wellbeing and recovery. The questionnaire was revised part way through collection and increased to 42 items, with additional questions on dizziness, wellbeing and recovery, sleep and sex drive. Sex drive questions were initially not included, in view of expert advice that the questionnaire should be comfortable to administer and discuss across a range of male athlete populations from different cultural backgrounds. After reviewing the initial results, however, questions on sex drive were added to improve sensitivity of the questionnaire. Both versions of the questionnaire included questions to provide demographic and athletic status information. Appendix A shows the initial version of the questionnaire prior to analysis, with questions added during the revision highlighted in red (version 1). Appendix A shows the final questionnaire (version 2) and associated scoring key, with sex drive being the sole section retained.

### 2.1. Internal Consistency and Reliability

To assess the performance of individual items and estimate reliability, a test–retest was performed. Forty-two male athletes were recruited from Australia, Norway and Sweden and received the LEAM-Q (Version 2) in either English, Norwegian or Swedish, as appropriate. The participants were asked to complete the questionnaire twice, 14 days apart. After the re-test, researchers asked the participants to identify any concerns they had with the items, including ease of understanding, relevance, and the appropriateness of the possible answers. Questionnaires were identified by subject number only and were collected either on paper or secure electronic format; Microsoft Forms or SurveyXact, (8200 Aarhus, Denmark).

### 2.2. Clinical Verification of Self-Reported Symptoms

A cohort of 352 male athletes was recruited for the main activity of the study, representing sports designated as weight sensitive (lightweight rowing, race walking, triathlon, road cycling, marathon, gymnastics, and ballet) or non-weight sensitive (openweight rowing, gymnastics, athletics, other). The LEAM-Q was completed online or on paper by all participants, with 42 ultimately removed due to missing key data. This resulted in 310 participants being involved in the final analysis of the LEAM-Q outcomes (Version 1: 183; Version 2: 127) against clinical assessment.

For the assessment of clinical markers, participants met at their respective test centre between 5 and 9 a.m. in a rested, fasted state (no food or fluid intake or prior physical activity on the morning of the assessment). Body mass was measured to the nearest 100 g and height to the nearest millimetre using calibrated instruments at the different centres. Body composition and BMD were assessed using Dual-energy X-ray absorptiometry (DXA) in the total body and site-specific modes on a narrow fan-beam DXA scanner (GE-Lunar Prodigy or iDXA, using GE enCORE analysis software version 15.0 or 16.2, Madison, WI, USA). Protocols included appropriate machine calibration and standardised positioning and were in keeping with best practice guidelines as previously described [76,77]. BMD was assessed for proximal femur and anterior posterior lumbar spine (L1–L4). For the Scandinavian cohorts the combined NHANES/Lunar reference database was used and for the Australian cohort the combined Lunar/Geelong as deemed most appropriate for the respective populations and low BMD was defined in Table 1.

RMR was measured either by metabolic cart (Oxycon Pro or Vyntus CPX, Jaeger GmbH, Hoechberg, Germany) or the first principles method [78] involving Douglas bags [78], depending on the testing location. All measures replicated participant preparation and presentation and were collected in a warm, quiet, and dimly lit room. For the metabolic cart method, a ventilated canopy hood system was used to assess RMR, with systems being calibrated before each test according to standards, and alcohol calibration weekly. Subjects rested for 15 min prior to collection. Oxygen consumption (VO_2_) and carbon dioxide production (VCO_2_) were assessed over a 30 min period and converted to kJ/min based on the Weir equation [79]. The last 20 min of measurement were used to assess RMR using the protocol defined by Compher et al. [80]. The first principles method replicated the processes as previously described [81]. To calculate the RMR_ratio_, the Cunningham (1980) equation [82] was used to calculate the predicted RMR of each subject: 500 + (22 × LBM [kg]). Resting Energy Expenditure (REE) was also calculated relative to fat free mass (FFM) as determined by DXA (kJ/kg FFM). As systematic differences were noted between the first principles and metabolic cart measurements and, as no threshold for RMR_ratio_ has been identified for male athletes, the lowest quartile of each method was used to indicate a “low RMR” finding (Table 1).

Blood pressure (BP) was obtained in a resting supine position using an electronic sphygmomanometer (Microlife BP A100, Widnau, Switzerland or HEM7320, Omron Healthcare, JA Davey Pty Ltd., Melbourne, VIC, Australia). The monitor was secured around the participant’s left upper arm, and automatically provided a reading of systolic and diastolic blood pressure, and resting heart rate.

Blood samples were collected within 30 min of completion of the RMR measurement, obtained via venepuncture from an antecubital forearm vein by a qualified phlebotomist. This ensured samples were fasted, rested and collected at a similar time of day for all participants [83]. For the Scandinavian cohorts, blood was clotted at room temperature for 30 min before being centrifuged at 1300× *g* for 10 min. Serum was transferred into tubes and stored at −80 °C until analyses. The serum from Kristiansand was analysed at St. Olavs Hospital (Trondheim, Norway) and serum from Oslo was analysed at Fürst medical laboratory (Oslo, Norway), for its content of glucose (CV 1.6%), insulin, cortisol (CV 3–5.4%), total testosterone (CV 6–9.2%), free triiodothyonine (T_3_) (CV 2.3–4.7%), and IGF-1 (CV4.8–7.5%). For the Australian cohort a single venous blood sample (2 × 8.5 mL serum separator tube) was used for the assessment of fasting IGF-1, cortisol, lipids, insulin, testosterone and T_3_ for analysis by chemiluminescent immunoassay through a commercial laboratory (Laverty Pathology, Bruce, ACT, Australia). IGF-1 was assayed using the DiaSorin Liason^®^ XL (DiaSorin Diagnostics, Sallugia, Italy, CV 2.5–6.4%), whilst cortisol (CV 2.9–5.2%), testosterone (CV 4.5–8.2%) and T_3_ (CV 2.6–5.3%) were assayed using the Siemens ADVIA Centaur XP (Siemens Healthcare Diagnostics Ltd., New York, NY, USA) as per the manufacturer’s recommendations. Fasting blood glucose levels were assessed via fingertip capillary sample using a portable meter and test strip (Accu-Chek^®^ Performa, Roche Diagnostics, Castle Hill, NSW, Australia, CV < 5%). Free testosterone was calculated from total testosterone, sex hormone binding globulin and albumin, or where unavailable 43 g/L, according to the method by Vermeulen et al. [84]. As the blood analyses were conducted at different laboratories, a “low” finding was determined using the lowest quartile of the reference range for the laboratory at which the measure was taken (Table 1).

### 2.3. Statistics

To assess the performance of the items and estimate reliability, the intraclass correlation coefficient (ICC) was used to calculate the difference between the test and the retest score using a two-way mixed random effects model.

The association between clinical outcomes and LEAM-Q variables were assessed including all subjects (*n* = 310), using multivariate linear or logistic regression models for all combinations of clinical outcomes (as responses) and screening variables from LEAM-Q (as predictors), including adjustment for age, BMI, elite athlete (yes/no) and centre (if there were data from multiple centres) (Table 2). In addition to the standard questionnaire scoring, a separate score was conducted for symptoms included in the EHMC [85]. This was assessed as a score for the sex drive questions “In general I would rate my sex drive as”, “Morning erections over the last month” and “How many morning erections compared to normal” in combination with the items “I feel tired from work or school”, “I feel lethargic”, “I feel strong and making good progress with my strength training”, “I feel very energetic in general”, I feel invigorated for training sessions and ready to perform well”, “I feel happy and on top of my life outside of sport”. Low sex drive was also categorised by using sex drive scores equal or greater than 2 on “Sex drive in general” or equal or greater than 2 on “The number of morning erections” and equal or greater than 1 for “Morning erections compared to normal” to represent reproductive dysfunction. Weight flux was defined by the difference between “highest” and “lowest body weight at current height” responses from the questionnaire.

ROC curves were used for evaluating, optimizing, and visualizing the performance of classifications of a continuous biomarker into two groups for predicting the clinical outcome of interest, LEA [86]. For LEAM-Q variables with significant association to one or more clinical outcomes (*p* < 0.05), optimal sensitivity was estimated using ROC curve analysis with Youden’s index [87]. At least 60% sensitivity was required to identify potentially useful screening variables, which were retained (Table 3). For clinical variables classified as “high” or “low”, this represented the test locations highest or lowest 25% percent of results, respectively (Table 1). Data were analysed using R (R Core Team 2020. R Foundation for Statistical Computing, Vienna, Austria) with the following extension packages, Hmisc [88] and pROC [89].

Given the recognised limitations of EA assessments in the field [5], LEA was operationally defined as having two or more primary indicators or three or more indicators overall. Primary indicators were derived from the male athlete triad [5] and secondary indicators from energy restriction and LEA literature [5,7,59,90,91,92]. After excluding 45 subjects missing at least three clinical markers, including at least one primary clinical marker, the remaining subjects (*n* = 265) were categorised as LEA-cases or controls by using the criteria outlined in Table 1. A two-sample *t*-test was used to analyse differences in the retained LEAM-Q variables between cases and controls (Table 4).

## 3. Results

### 3.1. Questionnaire Validation Process

Two items were removed from further analysis following the test–re-test process due to low ICC (“How would you describe your normal stool” and “I feel down and less happy that I used to feel or would like to feel”). Following this revision, fourteen-day test–re-test reliability ICC was 0.71. [59].

### 3.2. Subject Characteristics for Main Analysis

Of the 310 participants included in the analyses, 64% were elite athletes, 31% sub elite and 5% club level athletes from ten different countries. Half of these participants reported being full time athletes or professional, with 36% reporting placing within the top 10 at their respective international competition. Based on the definition summarised in Table 1, 2% of participants were classified as underweight, while none had low body fat levels, 24% had low BMD, 27% had low RMR, and low blood concentrations were found for testosterone (23%), T_3_ (17%) and insulin (26%). High blood cortisol concentrations were found in 28% of participants, while 30% had high LDL cholesterol. Meanwhile 2% of participants had hypoglycaemia and 11% had hypotension. Those who were underweight (*n* = 5) showed greater weight flux, lower T_3_, total testosterone, systolic BP, higher dizziness scores and less morning erections than the rest of the cohort (all *p* < 0.05). Those with hypotension showed no differences with any clinical variable or questionnaire score. Mean maximum oxygen uptake (VO_2max_) was 68.1 ± 7.2 mL/kg/min. Athletes from a weight sensitive sport had a lower height (179.8 ± 6.8 vs. 188.1 ± 7.3 cm, *p* < 0.001), body mass (71.1 ± 7.8 vs. 82.4 ± 12.7 kg, *p* < 0.001), BMI (21.9 ± 1.8 vs. 23.1 ± 2.4 kg/m^2,^
*p* < 0.001), FFM (62.4 ± 6.8 vs. 73.6 ± 9.6 kg, *p* < 0.001), RMR_ratio_ (0.98 ± 0.13 vs. 1.13 ± 0.13, *p* < 0.001), spine BMD Z score (−0.12 ± 0.97 vs. 0.41 ± 1.0. *p* < 0.001) systolic BP (117 ± 9.7 vs. 127 ± 10.3 mmHg, *p* < 0.001), and higher percent body fat (12.4 ± 3.8 vs. 10.1 ± 3.3%, *p* < 0.001) and T_3_ (5.4 ± 0.7 vs. 5.0 ± 0.97 pmol/L, *p* < 0.01) than those from non-weight sensitive sports. No trend was seen for a decline in free or total testosterone with increasing age.

### 3.3. Case Control Comparison

Forty-five subjects were removed from the classification into LEA-case or control based on incomplete clinical indicators, leaving 265 remaining subjects for this portion of the analysis (Table 4). Of these, 85 (32%) were classified as having LEA. LEA-cases were older, had a higher age of sport specialisation, lower spine and total femur BMD Z scores, systolic BP, RMR, total and free testosterone, free testosterone:cortisol ratio, IGF-1, T_3_, insulin and higher cortisol:insulin ratio, and total and HDL cholesterol compared to controls.

Sub section and total LEAM-Q scores were not different between LEA cases and control cohorts, with the exception of the sex drive score (Table 5). Of the 118 athletes answering the sex drive questions, 23.7% (*n* = 28) were categorised as having a low sex drive with lower total testosterone (18.0 ± 6.0 vs. 20.9 ± 5.6 nmol/L, *p* = 0.025), T_3_ (5.3 ± 0.7 vs. 5.6 ± 0.7 pmol/L, *p* = 0.047), and insulin levels (21.1 ± 10.3 vs. 25.8 ± 9.8 pmol/L, *p* = 0.045), lower femur BMD Z-score (−0.02 ± 0.97 vs. 0.39 ± 0.88, *p* = 0.041), and diastolic BP (64.7 ± 4.8 vs. 67.9 ± 7.6 mmHg, *p* = 0.044), while having a higher cortisol:insulin ratio (26.9 ± 17.4 vs. 20.6 ± 10.1, *p* = 0.035), and weight flux (10.2 ± 5.8 vs. 8.1 ± 4.1 kg, *p* = 0.037) compared with athletes with a normal sex drive. There was a non-significant trend towards lower free testosterone (0.8 ± 0.5 vs. 1.0 ± 0.4, *p* = 0.089), IGF-1 (29.4 ± 6.2 vs. 32.2 ± 7.1 nmol/L, *p* = 0.073), and testosterone:cortisol ratio (0.04 ± 0.02 vs. 0.05 ± 0.02, *p* = 0.074).

### 3.4. Utility of Clinical Variables

Table 6 describes differences in clinical and questionnaire variables between subjects classified as having low testosterone, RMR, T_3_, BMD and a high cortisol:insulin ratio compared to those having normal levels. Those classified as having low testosterone, RMR or T_3_ had a lower body mass, BMI, and systolic BP. FFM, total testosterone to cortisol ratio, free T_3_, systolic BP, and higher cortisol to insulin ratio. Those with low RMR_ratio_ were older, had lower height, body mass, BMI, FFM, systolic BP, total testosterone, IGF-1 and free T_3_ levels, and reported less frequent than normal morning erections. Those with low free T_3_ (*n* = 24) had lower RMR (kJ/kg FFM) and RMR_ratio_, free and total testosterone, free and total testosterone to cortisol ratio, IGF-1 and higher cortisol levels compared to their counterparts with normal free T_3_. Those with lower BMD showed no differences in key clinical markers of LEA compared to those with normal BMD. High LDL had no association with clinical markers thought to be indicative of LEA.

## 4. Discussion

Despite widespread interest, this is the first large scale attempt to validate a specific LEA screening tool for male athletes. Associations were seen between the LEAM-Q questions and clinical markers of LEA with adequate sensitivity in areas of dizziness, illness, wellbeing and fatigue and sex drive. Apart from sex drive, the developed questionnaire was, however, unable to distinguish between LEA cases or controls, as categorised by the researchers, for total score or any sub-score. This is an important finding given the number of questionnaires currently used to identify LEA in male athletes that are either validated only in females or not validated at all. Those classified as having low sex drive by the LEAM-Q questionnaire demonstrated multiple perturbations in clinical markers of LEA. A secondary finding was that perturbations in clinical markers of LEA tended to “cluster” but did not present uniformly across cases. The presentation of male athletes with LEA was different to characteristics shown in the literature on female athletes with LEA, both in the pattern of the questionnaire responses and the clinical markers.

Responses to the LEAM-Q questionnaire found several associations between sub-scores and perturbations in individual clinical markers. For example, sex drive was associated with total testosterone, T_3_, insulin and free testosterone:cortisol ratio, while weight flux was associated with cortisol:insulin ratio, dizziness was associated with glucose and insulin and insulin:cortisol ratio, illness was associated with T_3_, and wellbeing and fatigue were associated with high total cholesterol.

The LEAF-Q for LEA in females found an association between gastrointestinal symptoms and characterized LEA [59]. In contrast, the male participants categorised as cases in the present study did not have higher gastrointestinal scores than the controls, although participants with low T_3_ and low spine Z-scores did have higher scores. The physiological basis for an association between gastrointestinal symptoms and BMD is unclear. Gastrointestinal symptoms have been previously associated with self-reported exercise dependence and disordered eating scores in male athletes [66], and in male eating disorder populations [93]. Although there is a possibility of a sex-difference, gastrointestinal symptoms may also be more linked to the athlete’s sport type. Indeed, a mixed sport cohort of female athletes did not show links between gastrointestinal symptoms and LEA [94] previously reported in the LEAF-Q validation in endurance and weight sensitive sports [59].

Our study failed to show an association between clinical variables and questions around sleep or thermoregulation, and further research on these themes seems less likely to be productive. Although injury scores were associated with several of our biomarkers of LEA (Table 2), the sensitivity of these scores was low. Indeed, unlike the LEAF-Q validation and other studies in female athletes [59,95], our study failed to find an association between injury scores and BMD [66]. Typically, studies in both male and female endurance athletes have found correlations between bone stress injury rates and BMD [96], with one investigation of male athletes reporting that a cumulative risk assessment score incorporating both LEA and BMD [25] was predictive for bone stress injuries [25]. However, we note the lack of association between LEA and injury in a large scale, mixed sport female population [49] and suggest that in studies involving a diversity of sports, such as the present investigation, injury causation is likely to be multifactorial and less tightly related to LEA. It is possible that more targeted questions around injury within a uniform athlete group may improve the sensitivity of this factor in predicting LEA, but this would also reduce the applicability of the questionnaire across sports as is noted for the LEAF-Q [94]. Failure to find relationships between BMD, LEAM-Q questions and other markers of LEA in the current cohort may be due to the disassociation between acute measurements and the chronic nature of bone health [97,98].

Questions around dizziness were included in the LEAM-Q battery although they were removed from the LEAF-Q when the validation process found an association only with disordered eating rather than measured LEA [59]. In the present study, we found that adverse dizziness scores were associated with higher cortisol:insulin ratio and lower glucose and insulin. As there was no screening for disordered eating in the current validation, it is not possible to determine whether this was a sign of LEA or DE, and this limitation is acknowledged.

Higher illness scores were associated with lower T_3_ among our participants. Although this is in keeping with the findings of studies involving menstrual dysfunction [99], LEAF-Q scores [64] and participation in leanness sports [39], no association between illness and markers of LEA was seen in a large-scale mixed sport cohort [49]. Further research is required to understand the interaction between the immune system and EA in athlete populations. Indeed, a recent review of the complex relationship between nutrition and immune tolerance/resistance has recently proposed that energy restriction per se may not increase illness risk, and that previous associations reported in studies of athletic populations may be mediated by a common co-morbidity such as higher ratings of psychological stress [100]. Indeed, one study has reported an apparent disconnect between EA and the occurrence of upper respiratory infections in athletes who commenced high-intensity interval training [101]. Further research on this theme is warranted.

Other unexpected findings in the present study include the association between poorer wellbeing and recovery ratings and higher total, but not LDL, cholesterol. The reasons for this association are unclear and worthy of further investigation to identify whether this is a repeatable association and the possible underlying mechanisms. Furthermore, athletes in weight sensitive sports were noted to have higher body fat than those from non-weight sensitive sports. It is possible that this is due to perturbations previously observed in some groups assessed as being exposed to LEA [29] or poor within-day energy balance [102].

The clinical indicators most often associated with adverse questionnaire responses in our participants and the differentiation between LEA cases and controls were testosterone, cortisol, insulin, cortisol:insulin ratio, T_3_ and RMR. This is supported by other studies on LEA, within-day energy balance or energy restriction in males [14,58,96,97]. These markers may be most helpful in studying LEA in male athlete populations. Raised LDL cholesterol was associated with other clinical markers in the current study, but none fit the pattern expected with LEA. Further investigations of interactions between cholesterol metabolism and LEA or coincidental metabolic impairments are warranted, noting that LDL cholesterol is higher in anorexia nervosa patients than controls [103].

Overall, we found that LEA in a field setting is difficult to characterize with errors of measurement compounded by differences in the presentation of acute and chronic changes in clinical markers and individual differences in presentation [48]. Indeed, while we found an overlap in clinical presentations, there was also a divergence (Table 5) in both the clinical markers and the questions showing perturbations. Our results further highlight the folly of previous approaches to screening for LEA in male populations, including the use of the LEAF-Q from which questions on menstrual function have been excluded [67] or replaced with male reproductive questions [66], or those based on adaptations of female specific questionnaires that have not been validated for males [25,26].

The LEAF-Q was founded on the female athlete triad, associating questions on injury with low BMD, gastrointestinal dysfunction with LEA and the menstrual function score with clinically verified menstrual dysfunction [59]. In the current LEAM-Q validation, however, neither injury nor gastrointestinal symptoms were associated with LEA biomarkers with adequate sensitivity and were excluded from the questionnaire. The lack of utility of questionnaires developed for female populations in male cohorts is not unique to LEA; researchers have identified flaws in the application of female-derived surveys of disordered eating and body image [104,105,106] and have noted erroneous outcomes in clinical and research activities in other areas due to the use of poor screening tools [107].

The inclusion of the sex-drive variable in the updated version of the LEAM-Q warrants several comments. It was included as a proxy marker of reproductive function, to mimic questions around the menstrual cycle included in the LEAF-Q. It was not included in the first version of the LEAM-Q, due to external advice that it is challenging to obtain accurate information on sex drive given the possibility of stigma or embarrassment around admitting low sex drive or reduced morning erections. Furthermore, the accuracy of self-reports of sex drive has not been established. Nevertheless, subsequent discussion among the research team considering growing recognition of endocrine changes in male athletes associated with LEA [3,108] increased our interest in collecting information on sex drive within the LEAM-Q. Despite the caveats around such self-reported information, accuracy of recall over the last month, and the relatively smaller sample size in the analysis of this factor, we found perturbations to sex drive to be the most consistent indicator of LEA in male athletes, being the only questionnaire metric that differed between cases and controls. Further investigation is warranted in both males and females; indeed, it may be useful to interrogate sex drive in female populations as an adjunct to information on menstrual function or to address situations where the use of hormonal contraceptives interferes with an assessment of menstrual status. Indeed, females with anorexia nervosa are reported to experience a lower sex drive [109].

We were deliberate in designing our study to investigate a collection of biomarkers of LEA rather than assessing EA in each participant based on information on energy intake, exercise energy expenditure and FFM. We note both the lack of a standard methodology for EA assessment and the errors involved in estimating each of these components [48]. These issues, as well as the disconnect between an acute assessment and chronic time-course, over which an energy mismatch might have occurred, explain the conflicting outcomes of EA assessments and biomarkers of LEA in many studies [110]. No single marker is successful in identifying LEA; exposure may be best identified from a cluster of symptoms and with the exclusion of a differential diagnosis for some factors [111,112]. For example, Rogers and colleagues found that while 80% of an athlete cohort showed one or more of the possible symptoms associated with RED-S, only 11% recorded a low RMR [63]. Meanwhile, Stenqvist et al. identified male athletes with low RMR in the absence of any markers of LEA, including effects on BMD [97].

Although the best possible effort was made to characterise the clinical markers identifying LEA in the present study, further research is required to better identify thresholds indicative of perturbation in male athletes. In this study, the lowest or highest quartile was used for several variables where sub-clinical deficiency is likely to be important, but reference ranges for the marker are not yet available. Consistency in these cut-points will be important for future research and it is encouraging to see this develop for testosterone [5]. Ratios of cortisol:insulin and free testosterone:cortisol were significantly different between LEA cases and controls in our study; however, inconsistency of measurement units in previous research makes comparisons or the development of normative ranges challenging. While the overall data set was relatively large, key variables such as insulin, testosterone, cortisol and sex drive questions were only included in version 2 of the study and, as such, the sample size is much smaller for these key areas.

Previous research has shown that male athletes with higher exercise energy expenditure have lower EA [113] and males with eating disorders are more likely to have a focus on exercise rather than diet as a weight loss strategy [93]. Questions around training load and intensity have been successful in identifying male athletes with low testosterone [68] and exercise dependence with low testosterone cortisol ratio and high cortisol insulin ratio [24]. The LEAM-Q included a question on training hours, which was associated with aspects of sex drive. Given the diversity of the sports included in this investigation, this question was inadequate to capture differences in training load and the further development of questions of this nature may be worthwhile and have been included in the amended version of the LEAM-Q questionnaire.

A possible limitation of the current study was that, by nature, the multicentre, multi-country data collection resulted in multiple DXA machines, technicians and reference populations being used for assessment. Similarly, RMR was measured variously by a first principles and metabolic cart method and blood analysis was undertaken by multiple laboratories. Whilst these differences are acknowledged, the potential impact was minimised by using best practice protocols for data collection and using the lowest quartile for the testing site at which it was collected. Furthermore, the small differences in estimates of FFM and subsequent interpretation of RMR would likely be negligible.

The difficulty in validating this screening questionnaire may be due, in part, to the difficulty of identifying LEA in males and/or the need for further development of target questions. The specificity of key issues within certain sports or events is also recognised, meaning that although a questionnaire may successfully identify risk factors in a homogenous group, it may be less sensitive or play an alternative role in a different group or mixed population. For example, Rogers et al. found that the LEAF-Q, validated in endurance and weight-sensitive athletes, was able to “rule out” those at low risk of LEA in a mixed population of female athletes, while those scoring above the designated threshold would require further clinical assessment to identify LEA [96]. Indeed, while sex drive successfully differentiated between LEA cases and controls in the current study, it has also been used as a proxy for EHMC [68,70] and for disordered eating and exercise dependence [66]. Whilst these conditions are interrelated, a screening questionnaire can only act as a flag for further clinical assessment and not for diagnosis. It is noted that perturbations in testosterone and sex drive have been considered markers for EHMC, but in this study they were also associated with other endocrine and metabolic perturbations, highlighting the need for clarification of the interplay between LEA and EHMC.

This study provides unique information on the expression of LEA in a large group of male athletes across a range of sports and highlights the importance of asking about sex drive when screening male athletes for RED-S. It also confirms the need for sex-specific, sport-specific and, perhaps, calibre-specific screening tools in athlete populations. The LEAM-Q developed for the current study failed to clearly distinguish between athletes considered to be LEA cases and their control counterparts, with only the sex-drive sub-section having this utility. Nevertheless, it provides a bank of content-validated questions that could be of use for future studies in different populations. Further work from our group will focus on a new version of the questionnaire that extends the investigation of sex drive, with the addition of information on flux of body mass/composition and training load.

## Figures and Tables

**Table 1 nutrients-14-01873-t001:** Definition of Clinical Indicators of LEA.

Primary Indicators	Secondary Indicators
**Low T_3_:** lowest quartile * (<3.5 pmmol/L) [5]. **Low total or free testosterone:** Lowest quartile (<16 nmol/L, <333 pmol/L, respectively) [5]. **Low BMD:** Z-score <−1 for either AP spine or proximal femur [5,7]. **Low body weight:** Body Mass Index (BMI) <18.5 kg/m^2^ [5,91].	**Low RMR_ratio_** [5] lowest quartile for the testing method (<1.11 for first principles and <0.88 for metabolic cart measures). **Hypotension:** <90 mmHg systolic and/or diastolic <60 mmHg [59]. **Low body fat:** <5% as measured by DXA [92]. **Low IGF-1:** lowest quartile of the age dependent reference range at the testing site [90]. **High LDL cholesterol** (>3mmol/L) [59]. **High cortisol** (>550 nmol/L) or cortisol (nmol/l) insulin ratio (pmol/l) (>26.6).

Subjects were categorized as LEA if they had two or more primary indicators or three or more indicators overall. * “Lowest quartile” refers to the lowest quartile of the reference range at the specific testing site where the measure was taken.

**Table 2 nutrients-14-01873-t002:** Multivariate analysis of questionnaire items and associated clinical markers.

Questionnaire Item	Clinical Variable	N	Estimated Slope	SE	*p*-Value
Section 1: Dizziness					
1. Dizziness score	Glucose	264	−0.075	0.032	0.018
	Low insulin	117	0.600	0.227	0.008
	Proximal Femur BMD Z-score	302	−0.196	0.063	0.002
	High cortisol:insulin ratio	95	0.513	0.219	0.019
Section 2: Gastrointestinal Score					
2. Gastrointestinal score	AP Spine BMD Z-score	304	−0.136	0.0394	0.004
	Proximal Femur Z-score	302	−0.078	0.039	0.046
Section 3: Thermoregulation- no findings					
Section 4: Injury and illness					
4A How many acute injuries?	Low T_3_	177	0.683	0.279	0.014
	T_3_	177	−0.140	0.059	0.019
4B How many overload injuries?	Low T_3_	177	0.537	0.230	0.020
	T_3_	177	−0.130	0.054	0.018
	High cortisol	207	0.391	0.190	0.039
	High cortisol:insulin ratio	95	0.506	0.238	0.034
4D How many breaks in training have you had for acute injury?	High cortisol	209	0.389	0.168	0.021
	Cortisol	209	22.725	9.856	0.022
4F Number of days unable to train due to illness	Low T_3_	176	0.762	0.267	0.004
	T_3_	176	−0.191	0.054	0.001
4 Injury and illness score	Low T_3_	177	0.173	0.065	0.008
	T_3_	177	−0.038	0.014	0.008
	High cortisol	217	0.093	0.045	0.040
Section 5: Wellbeing and recovery					
5A Fatigue sub score	Total cholesterol	241	0.048	0.022	0.028
5D Poor recovery sub score	Total cholesterol	241	0.078	0.031	0.013
5E Low energy levels	Low insulin	117	0.2133	0.098	0.030
5 Poor wellbeing score	Total cholesterol	241	0.016	0.006	0.013
Section 6: Sex Drive					
6A How would you rate your sex drive in general?	High cortisol:insulin ratio	95	0.767	0.373	0.039
	Weight flux	115	1.908	0.579	0.001
	Training amount	114	7.995	4.010	0.049
	Low insulin	95	1.177	0.416	0.005
	Cortisol:insulin ratio	95	4.959	1.897	0.011
	Total Testosterone	115	−1.882	0.826	0.025
	Proximal femur BMD Z-score	112	−0.326	0.130	0.014
	T_3_	114	−0.195	0.090	0.033
6B How would you rate it over the last month compared to normal?	T_3_	114	−0.221	0.106	0.039
	Glucose	107	−0.172	0.077	0.027
	Low insulin	95	0.817	0.398	0.040
6C How often would you wake with a morning erection?	AP Spine BMD Z-score	115	−0.177	0.074	0.019
	Training amount	114	4.734	2.265	0.039
	Low free testosterone:cortisol ratio	114	0.4346	0.1946	0.026
	Proximal femur BMD Z-score	112	−0.228	0.073	0.002
	Low BMD	115	0.520	0.211	0.014
6D Over the last month how does the number of morning erections compare to normal for you?	Low RMR_ratio_	115	0.743	0.343	0.030
Low sex drive score	High cortisol:insulin ratio	95	0.206	0.103	0.045
	Weight flux	115	0.4819	0.180	0.009
	Low insulin	95	0.209	0.105	0.045
	Proximal femur BMD Z-score	112	−0.121	0.039	0.003
	Testosterone	115	−0.5874	0.2527	0.022
	T_3_	114	−0.074	0.028	0.009
Exercise Hypogonadal Male Condition	Weight flux	118	2.049	0.887	0.023
	Proximal femur BMD Z-score	115	−0.397	0.193	0.042

Significance set at *p* < 0.05, *n* = 310. “High” represents the top and “low” represents the bottom quartile of the test locations’ clinical variables, respectively.

**Table 3 nutrients-14-01873-t003:** ROC analysis including all subjects (*n* = 310) showing questionnaire items associated with clinical variables according to the multivariate analysis (Table 1) with a sensitivity of >60%.

Questionnaire Item	Associated Clinical Variable	Score Threshold	Sensitivity (%)	Specificity (%)
1 Dizziness score	High cortisol:insulin ratio	0.5	70	52
	Glucose	0.5	62	49
	Low insulin	0.5	70	54
4F Illness score	Low T_3_	0.5	64	46
	T_3_	0.5	67	47
5 Poor wellbeing score	Total cholesterol	19.5	61	56
5A Fatigue	Total cholesterol	2.5	82	31
6 Low sex drive score	T_3_	1.5	64	86
	Low insulin	0.5	96	28
	Weight flux	0.5	81	24
6A Sex drive in general	Total testosterone	0.5	87	26
	Weight flux	1.5	69	56
6B Sex drive over the last month	T_3_	2.0	71	98
6C Morning erections	Low free testosterone:cortisol ratio	0.5	63	57

**Table 4 nutrients-14-01873-t004:** Subject characteristics LEA cases vs. controls.

Variable	All (*n* = 310)	Controls (*n* = 180)	LEA-Cases (*n* = 85)	*p*-Value
Age (years)	27.9 ± 6.9	27.0 ± 6.7	31.2 ± 7.6	<0.0001
Age at specialization (years)	18.1 ± 7.7 ^(*n* = 303)^	17.9 ± 7.1 ^(*n* = 177)^	21.3 ± 8.6 ^(*n* = 77)^	0.0010
Height (cm)	181.6 ± 7.7	182.1 ± 8.4	180.5 ± 6.5	0.1232
Body mass (kg)	73.4 ± 10.1	74.9 ± 11.0	72.1 ± 9.3	0.0449
BMI (kg/m^2^)	22.2 ± 2.0	22.5 ± 2.0	22.1 ± 2.1	0.1256
Weight flux (max min weight)	9.1 ± 9.5	8.9 ± 5.7	10.1 ± 6.5	0.1390
VO_2max_ (mL/kg/min)	68.1±7.2	67.9 ± 7.1 ^(*n* = 129)^	67.9 ± 7.4 ^(*n* = 71)^	0.9369
DXA body fat %	11.9 ± 3.8	12.5 ± 3.5	12.3 ± 3.7	0.6941
DXA FFM (kg)	64.9 ± 8.7	65.7 ± 9.7	63.7 ± 7.6	0.1050
AP Spine BMD Z-score	−0.01 ± 1.00 ^(*n* = 259)^	0.05 ± 1.03 ^(*n* = 174)^	−0.28 ± 1.01	0.0147
Proximal Femur BMD Z-score	0.35 ± 1.0 ^(*n* = 257)^	0.31 ± 0.96 ^(*n* = 173)^	0.04 ± 0.92 ^(*n* = 84)^	0.0325
BP systolic (mmHg)	118.6 ± 10.4 ^(*n* = 247)^	119.9 ± 10.7 ^(*n* = 149)^	116.9 ± 9.7 ^(*n* = 76)^	0.0373
BP diastolic (mmHg)	67.6 ± 7.6 ^(*n* = 247)^	68.1 ± 6.5 ^(*n* = 149)^	67.3 ± 6.5 ^(*n* = 149)^	0.4088
RMR (kJ/kg FFM)	125.7 ± 16.3 ^(*n* = 286)^	130.8 ± 15.1	120.1 ± 14.9 ^(*n* = 82)^	<0.0001
RMR_ratio_	1.01 ± 0.13 ^(*n* = 288)^	1.05 ± 0.12	0.95 ± 0.12 ^(*n* = 83)^	<0.0001
Total testosterone (nmol/L)	19.8 ± 5.8 ^(*n* = 256)^	21.2 ± 5.5 ^(*n* = 168)^	17.3 ± 5.5 ^(*n* = 83)^	<0.0001
Free testosterone (pmol/L)	425.3 ± 139.1 ^(*n* =207)^	456.4 ± 136.2	383.7 ± 136.8	0.0008
Free testosterone:cortisol ratio	1.01 ± 0.47 ^(*n* = 199)^	1.10 ± 0.46 ^(*n* = 127)^	0.87 ± 0.43 ^(*n* = 727)^	0.0006
Total testosterone:cortisol ratio	0.05 ± 0.02 ^(*n* = 217)^	0.05 ± 0.02 ^(*n* = 139)^	0.04 ± 0.02 ^(*n* = 78)^	0.0002
IGF-1 (nmol/L)	28.7 ± 8.5 ^(*n* = 218)^	31.5 ± 8.3 ^(*n* = 123)^	24.8 ± 7.5 ^(*n* = 75)^	<0.0001
T_3_ (pmol/L)	5.3 ± 0.8 ^(*n* = 177)^	5.7 ± 0.5 ^(*n* = 104)^	4.9 ± 0.7 ^(*n* = 53)^	<0.0001
Cortisol (nmol/L)	461.5 ± 127.5 ^(*n* = 217)^	449.0 ±121.9 ^(*n* = 139)^	483.9 ± 134.7 ^(*n* = 78)^	0.0523
Insulin (pmol/L)	24.2 ±10.3 ^(*n* = 117)^	26.4 ± 10.9 ^(*n* = 61)^	20.8 ± 7.4 ^(*n* = 36)^	0.0079
Cortisol:insulin ratio	22.1 ± 14.5 ^(*n* = 95)^	19.3 ± 10.1 ^(*n* = 61)^	27.1 ± 14.7 ^(*n* = 34)^	0.0031
Blood glucose (mmol/L)	5.0 ± 0.4 ^(*n* = 264)^	5.1 ± 0.5 ^(*n* = 168)^	4.9 ± 0.5 ^(*n* = 74)^	0.0893
Total cholesterol (mmol/L)	4.6 ± 0.9 ^(*n* = 241)^	4.5 ± 0.8 ^(*n* = 159)^	4.8 ± 0.9 ^(*n* = 80)^	0.0292
LDL (mmol/L)	2.7 ± 0.8 ^(*n* = 239)^	2.7 ± 0.7 ^(*n* = 159)^	2.9 ± 0.8 ^(*n* = 78)^	0.0680
HDL (mmol/L)	1.5 ± 0.3 ^(*n* = 240)^	1.4 ± 0.3 ^(*n* = 159)^	1.5 ± 0.4 ^(*n* = 79)^	0.0303
Triglycerides (mmol/L)	0.9 ± 0.3 ^(*n* = 241)^	0.94 ± 0.34 ^(*n* = 159)^	0.89 ± 0.37 ^(*n* = 80)^	0.2783

Data are expressed as mean ± standard deviation, significance set at *p* < 0.05.

**Table 5 nutrients-14-01873-t005:** Variable scores in LEA cases and controls.

Questionnaire Item	Control (*n* = 180)	LEA Case (*n* = 85)	*p*-Value
1 Dizziness score *	0.8 ± 0.8	0.8 ± 1.0	0.7738
4F Illness score *	0.92 ± 0.98	0.76 ± 0.91	0.1997
5A Fatigue score *	4.48 ± 2.74	3.84 ± 2.76	0.0764
5 Wellbeing score *	18.71 ± 10.89	20.37 ± 10.32	0.2308
6 Low sex drive score *	1.96 ± 1.93 ^(*n* = 77)^	3.00 ± 2.51 ^(*n* = 38)^	0.0160
6A Sex drive in general *	0.86 ± 0.58	1.11 ± 0.80	0.0599
6B Sex drive over the last month *	0.17 ± 0.47	0.32 ± 0.34	0.1979
6C Morning erections *	0.75 ± 1.07	1.26 ± 1.33	0.0284
6D Over the last month how does the number of morning erections compare to normal for you? *	0.18 ± 0.62	0.32 ± 0.74	0.3102

* A higher score indicates a clinically less favourable presentation of symptoms.

**Table 6 nutrients-14-01873-t006:** Utility of clinical variables ^1^.

**Low Testosterone** **(*n* = 66)**	**Low RMR_ratio_** **(*n* = 71)**	**Low T_3_** **(*n* = 46)**	**Low IGF-1**	**High Cortisol** **(*n* = 60)**	**High Cortisol: Insulin Ratio (*n* = 27)**	**Low BMD** **(*n* = 63)**	**Underweight** **(*n* = 5)**	**High LDL** **(*n* = 73)**
**Physique and Clinical markers**								
**Lower** Height *, BM **, BMI **, FFM * F and T testosterone:cortisol ratio *** T_3_ *** Systolic BP * **Higher** HDL *	**Lower** Height ***, BM ***, BMI *, FFM *** T testosterone * T_3_ *** IGF-1 ** Systolic BP ** **Higher** Age *** BMD femur Z-score *	**Lower** BM **, BMI **, % body fat * F and T testosterone *** F testosterone:cortisol ratio *** Systolic and diastolic BP ** Diastolic BP * **Higher** HDL *	**Lower** RMR *** **Higher** Age *** Weight flux *** % body fat ** HDL *	**Lower** % body fat * F testosterone *** F and T testosterone:cortisol ratio *** T_3_ *** Cortisol:insulin ratio *** Total cholesterol *	**Lower** % body fat * F testosterone ** F and T testosterone:cortisol ratio ** T_3_ ** Glucose * **Higher** Weight flux *	**Lower** None	**Lower** T testosterone * Systolic BP ** **Higher** Weight flux *	**Lower** Cortisol ** **Higher** Age ** T testosterone:cortisol ratio * Total cholesterol *** TG **
						**Higher** TG **		
**Questionnaire scores ^2^**								
Higher poor recovery score * Lower injury and illness score *	Fewer morning erections compared to normal **	Lower general sex drive score *, lower GI score	Lower poor fitness score *** Lower fatigue score *** Lower Wellbeing score ***	Higher Injury and illness score *	Increased dizziness * Lower general sex drive *	None	Higher poor fitness score * Fewer morning erections compared to normal *** Higher dizziness score *	**None**

^1^ Definitions of “low” clinical markers defined in Table 1; ^2^ lower questionnaire score indicates a more normal response; higher scores suggest perturbations. * *p* < 0.05, ** *p* < 0.01, *** *p* < 0.001. RMR: resting metabolic rate; FFM: fat free mass; BP: blood pressure; BMD: bone mineral density; BM: body mass; BMI: body mass index; TG: triglyceride; F and T testosterone: free and total testosterone; HDL: high density lipoprotein; T_3:_ free triiodothyronine, IGF-1: insulin like growth factor one.

## Data Availability

The data presented in this study are available on request from the corresponding author.

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
