# Peer review of "Screening for Low Energy Availability in Male Athletes: Attempted Validation of LEAM-Q"

_nutrients, 2022, doi:10.3390/nu14091873_

Round 1

Reviewer 1 Report

This is a very interesting and extensive piece of work. However, I have concerns over the varied nature of many of the measures taken between centres. The authors have been honest and open from the outset and do not overstate conclusions. Furthermore, many of the findings may be useful in the field to inform the design of future questionnaires or monitoring strategies. However, the ability to apply these findings to different populations depends on the validity of the measures used, which is questionable.

The objective of the research was to screen for LEA... but EA was not directly measured. It is fair to note that measures of EEE and EI are notoriously inaccurate and unreliable, but you are then comparing questionnaire scores to indirect, secondary measure of LEA. There is a lack of clinical diagnoses. To compound potential error, ranks/quartiles are used instead of diagnostic thresholds, while measurements of RMR are taken with different techniques and are subject to systematic bias between laboratories. The use of DXA may also have varied between centres to affect the comparability of these measurements, while blood taking procedures/analyses also differed (reliability data and/or comparative data of the two serum separation methods on a common marker would be helpful).  It is impossible to know how much of the difference in RMR for example represents variation between samples and how much is due to differing techniques/equipment - the use of a correction factor cannot account for this uncertainty. The accuracy of LEA assessment is therefore sequentially (and exponentially?) impaired by 1) indirect measurements, 2) technical/measurement error, 3) a lack of numerical/scalar LEA criteria. We cannot know whether the bottom 25 %, or 5 % etc. etc. are suffering from LEA in this sample/population. Whilst the LEAF-Q also compared athletes in the top 50 % for EA with the bottom half across multiple variables, at least EA was directly measured. Furthermore, the ability of the questionnaire to identify LEA athletes was assessed by comparing to defined clinical outcomes such as eumenorrheic versus amenorrhieic athletes and high versus low BMD.

The inclusion of n = 5 recreational athletes is not justified and at odds with statements defining the sample as elite and sub-elite performers. Similarly, the inclusion of male athletes up to 50, in a study looking at changes in sex drive/inferred changes in sex hormones needs justifying from a physiological standpoint (considering hormonal changes with age). Reliability data for the various tests employed should also be provided. One issue worth addressing would be the use of the Accu-chek for capillary samples versus venous bloods for glucose. There have been some issues reported re reliability of this measure (e.g. Schifman, Nguyen & Page, 2014 - Reliability of Point-of-Care Capillary Blood Glucose Measurements in the Critical Value Range), while even the manufacturers state that accuracy (within 15%!) is observed in 95 % of samples. Finally, there have been many correlations and separate t-tests run. Have these been adjusted for multiple comparisons and how?

Another issue that stands out is the process used to identify athletes with LEA for comparisons with healthy controls (Table 1). Although a great idea in principle, I would be far more comfortable with this as a diagnostic/identification criteria if it had been previously validated (apologies if it has, but I can find no reference to such validation). Again, this validation would need to compare the decisions arrived at using your “two primary criteria, or three criteria in total” system with clinical diagnoses (e.g. ED, enforced time off training etc) or a valid, numerical indicator of LEA. The use of sex-drive questions (particularly those involving recalling the number of a.m. erections over the past month) could do with evidence on the reliability/validity of those particular measures regarding recall-error. Additionally, a citation to justify the RMRratio threshold for LEA is required.

More could have been made of current controversies RE the Male Triad. As far as I’m aware, this has not quite been universally accepted, with a lack of mechanistic evidence underpinning this model (despite evidence from cross sectional studies showing reduced BMD in male athletes). I’m unaware of evidence demonstrating the same relationship between LEA and oestrogen/oestradiol (although I’m probably not as well informed as you!). For example, despite the fact that Wassurfurth (2020) states “progesterone and oestradiol levels are reduced in response to LEA in male athletes [90]”, this citation actually refers to Sale & Sale’s 2019 review rather than direct evidence. And in fact, Sale and Sale conclude that “It remains to be clearly established whether there is or is not a male athlete triad and whether the bone health implications of reduced energy availability are seen at the same level as in females”. More could also have been made of the lack of/usefulness of specific biomarkers, specifically considering the rapid fluctuations/responses of many markers with feeding which limits their usefulness for indicating chronic LEA. Either by comparing to a specific criteria, diagnoses, or even reporting on the correlations between measures rather than with a specific indication of LE per se, the information on relationships between biomarkers could be very useful (e.g. the apparent utility of T3).

This reviewer would recommend that data could be presented in several smaller publications. For instance, the regression analysis reporting the relationships between markers could be extremely useful to indicate useful biomarkers for tracking proxy markers of LEA (such as T3 being associated with many markers such as T:C ratio, BMI, BP etc etc). Studies could be done in a smaller/controlled cohort to directly and accurately assess EA, and/or many of the physiological measures could be investigated as change scores The internal reliability (Cronbach’s alpha/Rasch analysis etc) could be reported. More detail of the validity and reliability of the sex-drive questions (e.g. comparisons with T:C ratio and/or other quantitative measures ,particularly considering the need for recall and age of the male athletes involved) could be evaluated. The “two primary criteria, or three criteria in total” system could be validated separately with clinical diagnoses (e.g. enforced time off training from a doctor, ED etc).

Responses to specific points/lines are given below:

L56         It would make sense to begin with the consequences and or prevalence of LEA rather than potential causes

L61         It would also seem sensible to being with the (more established) triad/LEA issues in females, before moving to more recent (and varied) observations in males. Both from the point of view of explaining physiological causes, recent interest/lacking data in males, and to form a narrative that considers the historical context

L91         More could have been of (lack of)/utility of specific biomarkers, specifically considering the rapid fluctuations/responses of many markers with feeding which limits their usefulness for indicating chronic LEA

L147       Why up to 50 years old, considering declines in testosterone which later become a main talking-point of the study?

L157       Suggest veering away from the term “mixed”, considering single sex cohort, and use of the word “mixed” in the statistical model. Varied competitive levels?

L172       Suggest not using “Items” (“Item’s”?) as a possessive noun. To assess the performance of individual items?

L216       Please justify/provide citation for RMRratio and justify threshold. However... considering the fact that 1) DXA was used with two separate set-ups/operators etc., then 2) you process data further to make a ratio, 3) RMR was calculated from two separate methods, 4) you order according to rank, there is a sequential loss of information with each step that makes me seriously question the validity of this approach. Are there any athletes/a sub population who tested in both conditions to isolate the amount of different to technical, rather than group differences?

L230-248              Can you please justify the use of two separate methods for obtaining plasma? I seem to remember (although not an active biochemist) that serum separator tubes may affect the quantification of certain proteins. Can you 1) provide justification/evidence that none of yours would have been be affected? 2) Provide reliability data for your assays?

Again... considering the lack of defined diagnostic criteria (and the fact that many of these markers are relevant in terms of deviation from an individual’s, baseline), is the use of quartiles/ranking valid

L250       “Items” again (as previous)

L252      Could the two-Way mixed random effects model be described any more? Is the random effect the different intercept per athlete/individual responses?

L260      Are issues over reliability/validity of recall over the last month acknowledged (how many erections over 30 days? When you’re asking an athlete to retrospectively remember instances where they’re only just becoming conscious...)? Have these been acknowledged/quantified previously?

L270       The abbreviation ROC doesn’t appear to have been defined in the paper at this point

L271       Again... can you say LEA is defined as a clinical outcome if it’s not directly measured?

L282       Multiple instances of multiple tests being run simultaneously. Are these adjusted for multiple comparisons?

Table 1. – L286 Thanks for the specific details and/or citations providing reference ranges. However, can you justify this approach for identifying LEA if there’s no direct assessment of LEA? Has this/similar been used previously?

L299       Why 5 % recreational athletes? These don’t seem to have been mentioned previously, or justified as warranting inclusion

L299-307              Sorry if I’m being slow. Aren’t many of the characteristics in this list only defined as “low” if in the bottom quartile? For these measures then, 1) why do many differ from 25 %? 2) If this is an artefact of rounding-up/down with separate groups... again, is this data useful? We know the lowest quartile is 25 %.

L316      Just checking... the weight-sports had higher body fat? Interesting, if so. Worth a discussion point? Does this affect the validity of either your assumptions or conclusions?

Author Response

The authors thank the reviewer for the time taken to understand the research and the detail with which you responded.  Responses to each point raised are included in the attached document and resulting amendments to the paper in the revised version.

Reviewer 2 Report

Congratulation for the wonderful paper. Thank you for this helpful contribution. I appreciate their methods including study design and data analysis. I write some comments below that could benefit the article.

Tables. Information is shown in duplicate. If it appears in the table, delete the information from the text.

Conclusion??? Conclusion is a main section and it must respond to the main aim of the study. It is advisable that conclusion be clear and concise. Therefore, it is not recommended that its length be greater than one paragraph.

Congratulation again!!!

Author Response

The authors thank the reviewer for your comments.  Responses to the specific points raised are in the attached file.
